# Nurses’ Role in Obesity Management in Adults in Primary Healthcare Settings Worldwide: A Scoping Review

**DOI:** 10.3390/healthcare12171700

**Published:** 2024-08-26

**Authors:** Emilia Piwowarczyk, Maura MacPhee, Jo Howe

**Affiliations:** 1School of Nursing, University of British Columbia, Vancouver, BC V6T 2B5, Canada; emilia_piwowarczyk@bcit.ca; 2Pharmacy School, College of Health and Life Sciences, Aston University, Birmingham B4 7ET, UK; j.howe1@aston.ac.uk

**Keywords:** obesity, obesity management, adults, nursing, primary healthcare, scoping review

## Abstract

Obesity is a chronic, prevalent, and complex health condition that adversely impairs physical and mental health. The World Health Organization calls for integrating obesity care into existing chronic disease management programs within primary healthcare services. This scoping review aimed to examine registered nurses’ roles in the primary healthcare management of individuals with obesity. A scoping review was conducted using the Johanna Briggs Institute methodology framework. Thematic analysis was used to identify and categorize nurses’ roles in primary healthcare obesity management of adults. Of 1142 documents included in this review, 15 papers met the inclusion criteria. Thematic analysis yielded the following six themes representing nurses’ major roles: patient-centred care, patient assessments, therapeutic interventions, care management, patient education, and professional development. This review identified that the literature on nurses’ roles primarily described their focus on lifestyle interventions (mainly nutrition and physical activity), anthropometric measurements, health planning, goal setting, supportive care, monitoring progress, and arranging follow-up. The Discussion highlights the importance of determining RN knowledge gaps and biases. More research is required to determine the need for additional RN pre-and or post-education related to obesity as a complex chronic disease.

## 1. Introduction

Many noncommunicable diseases are multifactorial. One such disease is obesity, which is defined as “a prevalent, complex, progressive and relapsing chronic disease, characterized by abnormal or excessive body fat (adiposity), that impairs health” [1] (p. E875). The World Health Organization (WHO) [2] classifies overweight as a BMI ≥ 25 kg/m^2^ and obesity as ≥30 kg/m^2^. Of note is that the WHO classification system has limitations, such as its inability to measure body fatness associated with health risks accurately. More promising anthropomorphic indices, such as the body shape index, consider fat distribution associated with increased cardiometabolic health risks [3].

Globally, obesity has had a twofold increase since 1990, similar to other serious chronic diseases, such as diabetes, cardiovascular disease, and chronic respiratory disease, which have risen by 60%, 54%, and 39·8%, respectively [2,4,5,6,7]. Obesity is found to affect every single country across the world, and the number of people living with obesity is projected to rise from one billion people affected worldwide today to approximately two billion by 2035 [8]. People can have healthy lives at any size, but when excessive adiposity impacts health, the risk of developing physical and psychological ailments increases. Some of the physical complications linked to obesity are diabetes, cardiovascular disease, cancer, musculoskeletal disorders, and gastrointestinal disease [2,9,10,11,12,13]. Potential psychological complications may include depression, anxiety, self-esteem, and poor body image [14,15].

At the 75th World Health Assembly in 2022, the WHO [16] introduced the Acceleration Plan to Stop Obesity. One of the recommendations in the plan was to integrate obesity management within primary healthcare chronic disease management. The WHO [17] first identified primary healthcare as essential to “health for all” in the Alma-Ata Declaration of 1978. In the declaration, primary healthcare was proposed as the first point of contact for individuals across the lifespan. Primary healthcare is essential for addressing non-communicable diseases, such as obesity, because it provides ongoing long-term care for disease prevention, identification, treatment, and maintenance [18]. In addition, primary healthcare is community-based, ideally serviced by multidisciplinary teams that have been found to improve individuals’ health outcomes significantly, while reducing emergency department visits and the use of acute care services [19].

A review of 19 international overweight and obesity guidelines for primary healthcare found that most guidelines recommended that obesity should be managed as a chronic disease by multidisciplinary teams [20]. As part of multidisciplinary healthcare teams, registered nurses (RNs) are well placed to address obesity management, given their competencies in comprehensive chronic disease management, patient-centred care, appreciation of the broader social determinants of health, and collaborative care planning and delivery with multidisciplinary team members, individuals, and their supports (e.g., family members) [21,22,23]. RNs are one of the largest global healthcare workforces, making up around 50% of the global health workforce [24].

Despite the need for primary healthcare RNs to provide obesity management, research indicates that they tend to avoid weight-related conversations. One study found that only 42% of individuals with obesity recalled having an RN or doctor address the topic [25]. Similarly, in 2021, Bright et al. [26] found that less than 50% of RNs, midwives or health support workers felt comfortable initiating a conversation with patients about weight and related health behaviours. Some of the most common reasons for RNs to avoid conversations about obesity were the awkwardness of raising the topic and stigmatizing beliefs, such as weight is the responsibility of the individual and not the RN or doctor [27,28].

Many international obesity guidelines are designed for a range of general healthcare providers. These guidelines do not clearly specify the roles of specific team members, such as RNs. For example, a clinical practice guideline by Obesity Canada states that “clinical discretion should be used by all who adopt these recommendations” [1] (p. E876). Understanding RNs’ roles in primary healthcare obesity management is necessary to optimize their full scope and practice within a multidisciplinary team. A previous review that focused on the RN’s role in primary healthcare obesity management for adults (i.e., 19–64 years of age) was published in 2008 [28]. Since then, nurses’ clinical practice approaches to obesity pathophysiology and treatment approaches have been updated, such as pharmacotherapy [29]. Given the advances in obesity management and updated clinical guidelines, the aim of this research was to synthesize the current literature in an updated scoping review addressing RNs’ roles in adult obesity management in primary healthcare settings globally.

### Significance

The results of this scoping review can be used to further develop and/or refine nursing-specific guidelines, algorithms, and decision support tools to assist primary healthcare RNs’ evidence-based chronic disease management for adults with obesity.

## 2. Methods

### 2.1. Protocol

This scoping review on nursing’s role in obesity management was conducted using the Johanna Briggs Institute (JBI) methodology framework [30]. This review follows the Reporting Items for Systematic Reviews and Meta-Analyses Extension for Scoping Reviews (PRISMA-ScR) checklist [31]. A scoping review aims to broadly and systematically map out a comprehensive overview of the available literature on a topic [30]. Scoping reviews can use a variety of evidence, such as reviews, grey literature, and studies with different methodologies, to provide a broad summary of a topic [31].

### 2.2. Eligibility Criteria

Articles were eligible for inclusion if they were in English, peer-reviewed, and published in the last five years between 1 January 2018 and 1 January 2023. Given nurses’ access to new clinical guidelines for obesity management [29], a decision was made to limit the search to the past 5 years. Because of an unexpected time lag in completing this review, an additional search was conducted to include documents up to 30 April 2024. As per the JBI methodology framework, the inclusion and exclusion criteria for this review are described in Table 1.

### 2.3. Information Sources

The evidence sources were compiled from databases and citation searching. The information sources used were CINAHL (via EBSCOhost), MEDLINE (via EBSCOhost), and PsycInfo (via EBSCO, EBSCO Services, Ipswich, MA, USA) and included published peer-reviewed qualitative, quantitative, and mixed-method studies; reviews, such as systematic and scoping reviews; pictorial and educational articles; and published dissertations and theses. An academic medical information specialist assisted with the development of the search strategy.

### 2.4. Search Strategy

The keywords from the research question were used to construct an initial search strategy, beginning with the CINAHL database and the PubMed search engine. These were nursing, community, and obesity management. Search terms, adapted and refined through a preliminary search are shown in Table 2 for the following three databases: CINAHL, MEDLINE, and PsychInfo. The major headings and keywords were combined using Boolean “Or” to create unique concept blocks for each database. The full search strategy for each database can be found in Appendix A.

### 2.5. Selection of Sources of Evidence

The final search results were uploaded to Covidence© systematic review software (2023) to remove duplicates and screen titles and abstracts. One reviewer (EP) screened all the titles and abstracts, and a second reviewer (MM) completed a consensus check on a random selection of 10% of the abstracts. Any disagreements were resolved via discussion between the two reviewers. The final included documents were stored, managed, and cited using EndNote^TM^21 and thematically coded using NVivo^®^ 14.

### 2.6. Process of Data Extraction

An Excel spreadsheet was used for data extraction of all included documents based on the following descriptive characteristics: title, study design, population, setting, country, aim, and key findings. Appendix A contains all key descriptive characteristics.

### 2.7. Thematic Analysis

The following six phases of reflexive thematic analysis by Braun and Clarke [32] were used to guide the synthesis process: (1) become familiar with the data, (2) code (create concise labels), (3) create first themes, (4) develop and review themes, (5) refine, define and name the themes, and (6) produce a report. An inductive coding process was used to derive the six major themes pertaining to RNs’ roles in the obesity management of adults receiving primary healthcare in community settings.

## 3. Results

Of the 1142 documents identified in the search, 198 were excluded as duplicates, 884 were excluded after title and abstract screening, and 3 were not available for retrieval. After conducting a full-text screening of the remaining 57 documents, 11 met the inclusion criteria. Forward–backward citation searching from the 11 included documents identified 4 additional documents that met the inclusion criteria. A total of 15 documents met the inclusion criteria. The screening and selection process is detailed in Figure 1. The PRISMA checklist is included in Appendix A. This scoping review is registered under its title in Open Science Framework.

### 3.1. Characteristics of the Studies

Table 3 lists the characteristics of the 15 included documents [34,35,36,37,38,39,40,41,42,43,44,45,46,47,48]. The documents vary in methodology and are categorized according to the authors’ methodological descriptions as follows: six quantitative [38,39,43,45,47,48,49], three qualitative [35,40,41], two mixed-method [37,44], a review [34], a commentary [36], an educational article [42], and a protocol [46]. The documents also vary regarding their primary healthcare setting and geographical locations. The clinical settings were general practice offices [40,44,45,46], primary care clinics [35,37,48], an outpatient obesity clinic [34], and community care [36,38,41,42]. The community settings included the patients’ homes [36,41], a faith community setting (church) [42], and a municipal worksite [48]. Other included primary healthcare settings were remote telephone monitoring [43], a combination of telehealth and a general practice clinic [47], and no-contact multidisciplinary coordination and management [39]. Geographically, the ten countries represented in the included documents were Australia [40,44,45,46], Spain [38,39], Brazil [35,43], Canada [37], the USA [42], the United Kingdom (U.K.) [36], Italy [34], Finland [48], the Isle of Man [41], and India [47].

### 3.2. Synthesis of Results

The thematic analysis yielded the following six themes: patient-centred care, patient assessments, therapeutic interventions, care management, patient education, and professional development. The matrix used for all six themes is located in Appendix A. A narrative summary follows for each theme.

#### 3.2.1. Patient-Centred Care

The patient-centred care theme focuses on each individual’s specific health needs and therapeutic relationship building (aka relational practice). Nurses’ roles in patient-centred care were reported in ten of the documents [34,36,37,38,40,42,44,45,46,48]. Examples of patient-centred care included cultural awareness and adaptability [35,37,40,44,45]; non-stigmatizing, non-judgemental, and non-biased care [34,37,42]; therapeutic relationship building [34,35,38,43]; family engagement [34,39,42]; and respectful first conversations about living with obesity [36,37,42].

#### 3.2.2. Patient Assessments

All 15 documents discussed the importance of comprehensive nursing assessments including anthropometric measurements, lifestyle factors, physical health, mental health, and medical background. Anthropometric measurements were part of every nursing assessment [34,35,36,37,38,39,40,41,42,43,44,45,46,47,48]. The anthropometric measurements were basal metabolic index (BMI) [34,35,36,37,38,39,41,42,43,44,45,46,47,48], waist circumference [34,35,38,43,44,45,46], and weight [40]. Thirteen documents discussed the RN’s role in assessing lifestyle factors [34,37,38,39,40,41,42,43,45,46,47,48,50], such as nutrition [34,35,37,40,42,43,45,46,47,48], physical activity [34,39,43,45,46,47,48], social determinants of health [34,37,38,43,47,48], lifestyle behaviours [34,42,43,44,47,48], alcohol intake [34,39,43,48], smoking [39,43,47,48], sleep [34,47,48], readiness for change [34,37,41], quality of life [39,45,48], and social isolation [48]. Nutrition was the most assessed lifestyle factor, including nursing evaluation of previous diets and their effectiveness [34], current dietary practices [46], types of foods typically consumed [47], amount of healthy foods consumed [43,45,48], amount of fast foods consumed [47], eating patterns [43], meal frequency [34,42,47,48], amount of meals per day [43], portions per meal and portion size [34], hunger cues and rewards [42], emotional or stress eating [36,37], eating disorders [34], the potential for improved dietary changes based on food diary use [40], and adherence [34].

Physical health assessments were included in nine of the documents [34,35,37,38,39,43,44,46,48], and six of these documents included specifics about the management of physical co-morbidities [34,35,37,39,43,48], such as elevated blood pressure [34,35,38,45,46,48], cardiometabolic factors [34,38,39,46,48], chronic physical health risks [35,37,44,46], and chronic pain [39]. A mental health assessment was reported in five of the articles [34,36,37,39,48], including a nursing assessment of anxiety [34,37,39,48], depression [34,37,48], stress [34,36,48], body image [34,37], and psychotropic medication use [37]. The RN’s role in assessing past healthcare history was mentioned in three articles [34,36,48], which entailed obtaining a medical history [34,37] and a pharmaceutical history [34,48].

#### 3.2.3. Therapeutic Nursing Interventions

Therapeutic nursing interventions are nurse actions that provide holistic care that addresses each individual’s physical mental, spiritual, and social care needs. Delivery of therapeutic nursing interventions for individuals living with obesity included RNs’ roles as multi-disciplinary team members [34,35,37,38,39]. Other cited disciplines included nutritionists [38,39], dietitians [34,37], psychologists [34,38,39], mental health workers [37], physical activity teachers [35,38,39], and physicians [34,38,39,50]. Two documents described RNs as multidisciplinary team leaders and team coordinators [38,39].

Six nursing interventions for obesity management in primary healthcare involved delivering patient education [34,35,36,37,38,40,41,42,43,44,45,46,47,48], lifestyle behaviour change techniques designed to elicit change [34,35,36,37,38,40,42,43,44,45,46,47,48], supportive care to empower, support, and motivate [34,35,36,38,39,40,42,43,46], patient referrals to other services [35,41,42,44,45,46], and assessment/early identification of obesity-related complications [34,42,43,46,47]. Complications included cardiovascular disease [35,42,43,46,47], type 2 diabetes [35,42,43,46], metabolic disease [43,47], and falls from poor mobility [35]. Therapeutic nursing interventions also included motivational interviewing (MI) [34,41,44,47], which was used to elicit internal motivation in individuals for sustained long-term behaviour change and weight loss [34,47]. However, RNs reported a lack of confidence when using MI for obesity management [41,44].

Nursing therapeutic interventions for individuals living with obesity also focused on RNs’ use of the 5As framework [37,42,44,45,46]. There were two types of 5As frameworks reported in the reviewed documents. One 5As framework (assess, advise, agree, assist, and arrange) [42,44,45,46] is based on a behaviour change model initially used for smoking cessation and later adapted for obesity management [42,46,51]. The second 5As framework (ask, assess, advise, agree, and assist) is a step-wise method for obesity management for primary healthcare providers, developed by the Canadian Obesity Network (aka, Obesity Canada) [37,52].

The goals of care used to monitor intervention efficacy were reduced anthropometric measurements [34,35,38,39,40,41,42,43,45,47,48], changed lifestyle behaviours [34,35,37,40,42,43,44,45,46,47], prevention or improvement of physical co-morbidities [34,38,39,43,45,47], improved quality of life [37,39,45,46,47,48], achievement of patient-set goals [34,37,40,44], improved emotional well-being [39], and identification and care planning for identified root causes of obesity [34].

#### 3.2.4. Patient Education

Patient education topics covered by RNs included obesity as a chronic disease, including its etiology and pathophysiology [34,43]; the health risks associated with obesity [34,41,42,43,44,46]; and the importance of preventing and treating physical comorbidities [34,38,43]. Another educational topic was lifestyle factors related to obesity management, such as nutrition [40,41,42,43,44,46,47,48,49,50,51,52,53,54], physical activity/exercise [40,41,47,48,49,50,51,52,53], alcohol consumption [40,42], and smoking [40,52]. Nutrition was the most cited patient education topic provided by RNs including healthy nutrition [34,36,38,41,43,45,50]; healthy food choices and substitutions [34,40,43,47]; calorie counting [36,40,41,42,47]; nutritional goal-setting and advanced planning (before grocery shopping) [38,40,42,45,47]; awareness of hunger and satiety cues and ways to avoid emotional eating [37,42,47]; water intake [42,43,47]; food preparation tips [43]; identifying and removing/avoiding environmentally triggering foods, including takeout food [47]; food diary use [40], including systematic tracking of the frequency [43] and quantity [40] of food intake; and cultural preferences influencing food choices [37]. Nurses also shared information on potential lifestyle change interventions or supports including support groups [36,41], the use of anti-obesity medications [34,41,42], and metabolic surgery [35,41].

#### 3.2.5. Care Management

Nine documents discussed the specific obesity management factors that RNs were expected to monitor, including medication adherence [34], blood pressure [35], weight [35,40,41,43,44], BMI and waist circumference [44], progress towards behaviour change goal accomplishment [40,42,44,45,46], capacity to handle barriers or difficulties [42,44,46], and exercise regime [47].

In two articles, RNs were responsible for monitoring the delivery of care via multidisciplinary teams [38,39]. Four articles discussed obesity care monitoring via telephone calls [34,40,43,47]. This remote monitoring included encouragement and positive affirmation [40,43], clarifying doubts, and reinforcing evidence-based information [43]. Care management also included scheduling and follow-up for individuals [34,35,37,38,40,42,43,44,45,46,47], referral submissions [35,41,42,44,45,46], and coordination of care among team members [35,38,39,44].

#### 3.2.6. Professional Development

Nursing professional development topics associated with the care of individuals living with obesity included the following: reflexivity, continuous learning, and tools. Nursing reflexivity refers to RNs’ capacity to recognize their own biases and knowledge gaps [35,37,41]. Nurses’ use of reflexivity was associated with improved clinical judgment when caring for individuals living with obesity [42]. Nursing professional development included awareness and utilization of evidence-based guidelines for obesity management [35,36,43,44,46]. Four countries developed primary healthcare/community care guidelines for obesity management, such as the Australian guidelines for the management of overweight and obesity [46], the Brazilian obesity guidelines [35], “Management of Obesity” by the Scottish Intercollegiate Guidelines Network (SIGN) [36], and obesity guidelines from the National Institute for Health and Care Excellence (NICE) [36]. In addition to guidelines, one document discussed the importance of RN access to decision support tools or care pathways with evidence-based information on assessment, intervention planning, education, care monitoring, and evaluation [47].

## 4. Discussion

This scoping review mapped the current literature landscape and found the following six themes that capture the nurse’s role in managing adults with obesity in the primary healthcare setting: (1) patient-centred care, (2) patient assessments, (3) therapeutic interventions, (4) care management, (5) patient education, and (6) professional development. The nurses’ strengths in managing obesity were highlighted in this review, such as dietary assessment, anthropometric measurements (height, weight, BMI), personalized patient-centred care, goal setting and health planning, interventions for lifestyle behavioural changes (e.g., nutrition and physical activity education), monitoring, supportive care, and follow-up.

Nearly thirty years have passed since the WHO (1997) [53] expert consultation on obesity convened in Geneva, Switzerland, and recognized obesity as a disease. Since then, many countries and medical organizations have followed suit and updated their position statements to recognize obesity as a chronic disease [54,55,56,57,58,59,60,61]. Despite the momentum to redefine it in the last couple of decades, our review identified only four articles that defined obesity as a chronic disease [34,35,37,43]. The lack of definition updates in our review may be due to the ongoing debate over obesity as a chronic disease [55,57,62,63,64,65,66,67]. The Obesity Society (2008) [57] analyzed the pros and cons of a chronic disease definition and found more advantages than disadvantages. The advantages were identified as reduced weight stigma and discrimination, increased medical training, adding credibility to the field of obesity medicine, and better resources for obesity prevention, treatment, and research. Additionally, the American Medical Association in 2013 [58] determined that obesity meets the criteria of chronic disease. One study [37] in this review delivered an educational intervention on obesity as a chronic disease and reported non-statistically significant intervention outcomes (e.g., obesity-focused patient discussions). Another study by Aboueid et al. [66] found that RNs and other healthcare providers who viewed obesity as a chronic disease were likelier to engage in weight conversations with their patients. Overall, this review found that nearly half of the studies (n = 5/11 of varying designs) reported barriers that prevented RNs from engaging in obesity care, such as lack of confidence, knowledge, training, equipment, heavy workload, feeling embarrassed or awkward, fear of upsetting patients, and personal views of obesity care [35,37,40,41,44]. One study in this review [44] questioned whether the presence of these barriers negatively influences RNs’ capacity to fulfill their roles with respect to primary healthcare obesity programs. Chronic disease management is one of the RNs’ three main role responsibilities in primary healthcare [67], and RN-led chronic disease management has been effective at improving blood sugar control, blood pressure, disease symptoms, lifestyle improvements and patient satisfaction [68,69,70].

More research is needed to determine if barriers associated with RN obesity management require extra education or preparation (pre- and/or post-graduation) to deliver quality care for adults living with obesity and associated physical and mental co-morbidities. In baccalaureate nursing education, content on weight bias and stigma is included in theory, simulation, and practice-based education with promising outcomes (i.e., increased awareness of weight bias and strategies for addressing bias) [71,72]. Post-education, RNs can act as positive role models and mentors in situations where obesity bias is present but not acknowledged or addressed. For example, in one cross-sectional study with RNs, almost 50% of participants reported witnessing or providing decreased quality of care for patients with obesity. A primary reason for substandard care was the belief that individuals with obesity should do more to help themselves [73]. Proactive educational interventions, particularly in baccalaureate programs, hold promise with respect to new graduates’ more objective assessments and less subjective assumptions about patients with obesity.

A well-documented barrier to obesity care is weight bias and stigma in primary healthcare settings where negative outcomes have included intervention avoidance, shorter visits, fewer screenings, missed diagnoses, and patient reports of feeling disrespected, unheard, and unwelcome [14,74,75]. Three review documents [34,37,42] discussed the importance of non-judgemental and non-stigmatizing RN care and one [42] document discussed RNs’ use of self-reflection to identify their own biases. Discriminatory RN treatment was associated with stigmatizing beliefs, such as people with obesity are “lying about trying”, “unmotivated”, “unattractive,” and “lazy” [28,66,73,76]; some studies reported that RNs’ discriminatory behaviours were associated with non-evidence-based assumptions that weight is a matter of personal choice and self-control [73,77]. Ringel and Ditto [78] found that RNs’ assumptions about individuals’ weight controllability were significantly correlated with moral disapproval of obesity and feelings of disgust and body shaming.

Weight controllability has been associated with the “calorie in, calorie out”, or “eat less, move more” explanations of weight control [79]. Current evidence suggests that multiple factors influence how the body stores or releases energy [1,80,81]. Some factors that disrupt energy storage balance are genetics, medical conditions, medications, biological factors, psychosocial factors, behavioural and lifestyle factors, cultural factors, and environmental factors [1,82,83,84]. Torres-Carot et al. [79] emphasized how the “calorie in, calorie out” model does not capture the complex mechanisms of energy storage. Unfortunately, simplifications, such as “eat less, move more”, still challenge current evidence about obesity management. The literature includes a number of articles that suggest wide variability in RNs’ knowledge of obesity management [82,83,84]. One cross-sectional survey of RNs’ knowledge about obesity care reported that RNs’ correct answers to weight management questions ranged from 33 to 99% [82]. To understand if RNs and other healthcare providers use evidence-based or common knowledge for obesity management, one study [85] found that 39% of respondents predominantly relied on common knowledge while providing obesity care. Those providers with the least amount of obesity management knowledge were more likely to overestimate their understanding of obesity, and they were less likely to seek additional information because of over-confidence [85]. As mentioned previously, more research is needed to understand knowledge gaps in pre- and post-graduate education (for RNs and other healthcare providers).

Pre- and post-education of obesity management for RNs and other healthcare providers can be aided by evidence-based tools, such as the 5As framework of obesity management (ask, assess, advise, agree, and assist), which was adapted from a smoking cessation tool based on behaviour change theory [86]. The use of the tool by RNs needs further exploration, but one pre–post-intervention study with physicians reported a statistically significant uptake of evidence-based obesity care after 5As framework training. Significant outcomes included the completion of medical obesity assessments and individualized follow-up and planning [87]. The 4Ms (mental, mechanical, metabolic, and monetary) tool is also available to guide RNs’ comprehensive obesity management [88]. Each M stands for a category to assess, such as mental (i.e., mood, anxiety, knowledge, sleep), mechanical (i.e., pain, osteoarthritis, obstructive sleep apnea, lymphedema), metabolic (i.e., nutrition, diabetes, hypertension, dyslipidemia), and monetary (education, employment, health insurance and disability) [88]. In addition, since people with obesity experience weight stigma, which can be retriggered in healthcare settings, the trauma-informed care (TIC) approach may create safer spaces for them. The TIC approach recognizes that every person may have trauma, and difficult opening questions and conversations need to be welcoming, supportive and non-judgmental [89]. While these tools can help direct care, barriers, such as lack of confidence and perceived poor knowledge, indicate that obesity education may help enhance RNs’ (and others’) obesity management of adults in primary healthcare settings.

### Limitations

This scoping review covered a narrow time period of English-only documents. The focus was on RNs’ role in the primary healthcare management of people with obesity; however, RNs are integral members of teams, and the literature on team-based care for obesity management was not explored in this scoping review. In addition, primary healthcare covers the developmental continuum, and “when” and “how” to begin addressing obesity and its impact on health and well-being has not been well established. This review only focused on adults aged 18–64, excluding other individuals along the age spectrum. A decision was made to conduct a scoping review, given the exploratory nature of the review question and our desire to learn more about gaps between RNs’ current job roles and ongoing knowledge and skills advances in obesity management. The JBI scoping review framework was used to provide rigour and clarity to our process, but scoping review designs tend to be less rigorous than systematic reviews. It should also be noted that our review did not address differences that may exist within and across countries with respect to nurses’ scope of practice and capacity to regulate themselves as a professional body.

## Figures and Tables

**Figure 1 healthcare-12-01700-f001:**
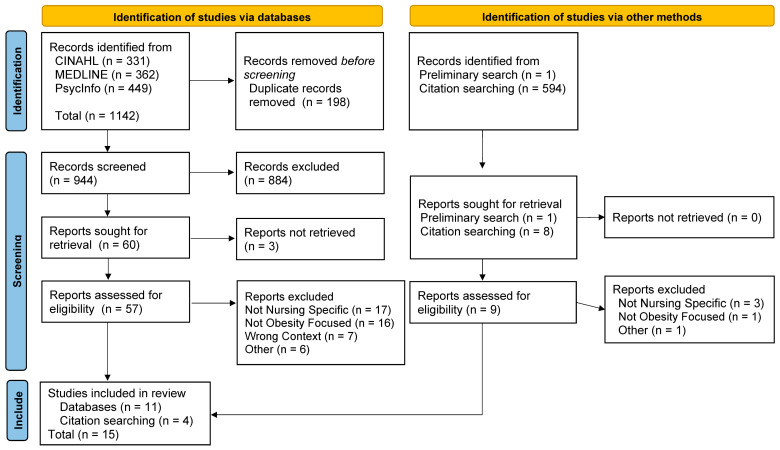
PRISMA-ScR Flow diagram of the selection of sources [33].

**Table 1 healthcare-12-01700-t001:** Inclusion and exclusion criteria.

	Inclusion	Exclusion
Population	Registered nurses Case management nurses General practice nurses	Student nurses Advanced practice nurses Nurse practitioner (NP) Clinical nurse specialist (CNS) Certified Registered Nurse Anesthetist (CRNA) Certified Nurse Midwife (CNM)
Concept	Individuals with obesity Adults (19–64 years old) Nurses’ role or intervention	RN’s role indecipherable from other disciplines Primary focus on other disease (e.g., cancer, serious mental illness, NAFLD, T2DM, etc.) Bariatric and pharmaceutical related interventions Population who requires expertise beyond scope of this paper (i.e., perinatal, postnatal, prenatal, and parenting and people with developmental, and intellectual disabilities)
Context	Worldwide Primary healthcare setting District nursing General practice Community settings (home care, occupational health, or faith-based)	Any setting outside inclusion criteria

Note. The inclusion and exclusion criteria for this review are contingent on the posed research question guided by the stipulated population, concept, and context (PCC) [30].

**Table 2 healthcare-12-01700-t002:** Search strategy.

	Medical Subject Headings (MeSHs) and Descriptors	String/ Boolean	Keywords (Title or Abstract)
Population	“nurses” OR “nursing” OR “Public Health Service Nurses”	Or	nurs*
	And		
Concept	“role” OR “Nursing Role” OR “Nursing Interventions” OR “Professional Role” OR “Delivery of Health Care” OR “Practice Patterns, Nurses” OR	Or	role* or “nurs* intervention*” OR “nurs* strateg*” OR “nurs* role” OR “nurs* guided” OR “nurse-directed” OR “nurse-led” OR “nurse-managed” OR “nurs* function*”
	And		
	“obesity” OR “Obesity, Morbid” OR “weight control” OR “Weight Reduction Programs” OR “weight management” OR “obesity management” OR “Body Weight Maintenance” OR “Body Weight Changes” OR “weight loss+” OR “Weight Reduction Programs+” OR “Body Weight” OR “overweight” OR “body mass index” OR “Obesity (Attitudes Toward)”	Or	obes* OR “high BMI” OR “high body mass index” OR “weight control” OR “weight reduction” OR “weight management” OR “overweight”
Context	Omitted (too restrictive)		Omitted (too restrictive)

**Table 3 healthcare-12-01700-t003:** Descriptive Characteristics of the Included Documents.

Frist Author/Year/Citation		Title	Study Design	Population	Setting	Country
Barrea (2021) [34]		The role of the nurse in the obesity clinic: a practical guideline.	Review	RNs caring for people living with obesity (PwO)	Outpatient obesity clinics	Italy
Braga (2020) [35]		Actions of nurses toward obesity in primary health care units.	Qualitative	Primary healthcare nurses (PHNs)	Primary healthcare Units	Brazil
Brewah (2018) [36]		Can community nurses take on obesity?	Commentary	District/ community RNs caring for homebound PwO	Home care	U.K.
Campbell- Scherer (2019) [37]		Changing provider behaviour to increase nurse visits for obesity in family practice: the 5As Team randomized controlled trial (RCT).	Mixed-methods, RCT, and qualitative	Chronic disease RNs in a primary care clinic providing care for PwO	Primary care network clinics	Canada
Fernández- Ruiz (2018) [38]		Short-medium-long-term efficacy of interdisciplinary intervention against overweight and obesity: randomized controlled clinical trial.	RCT	Multidisciplinary program for PwO led and coordinated by RNs	Community care centre	Spain
Fernández- Ruiz (2018) [39]		Impact of the I(2)AO(2) interdisciplinary program led by nursing on psychological comorbidity and quality of life: randomized controlled clinical trial.	RCT	Multidisciplinary program for PwO led and coordinated by RNs	Community care centre	Spain
Govindasamy (2023) [40]		Experiences of overweight and obese patients with diabetes and practice nurses during implementation of a brief weight management intervention in general practice settings serving culturally and linguistically diverse disadvantaged populations.	Qualitative	RNs PwO who are culturally and linguistically diverse with socioeconomic disadvantage	General practice office	Australia
Hinks (2022) [41]		Exploring community nurses’ views on the implementation of a local weight management pathway.	Qualitative	District and community RNs	Community care	Isle of Man
Kelley (2018) [42]		The role of the faith community nurse in weight management.	Opinion	Faith community RNs providing care for PwO	Faith community setting (church)	USA
Palmeira (2019) [43]		Effect of remote nursing monitoring on overweight in women: clinical trial.	RCT	RNs providing remote weight monitoring for PwO	Primary care Remote nursing	Brazil
Parker	2018 [46]	Preventing chronic disease in patients with low health literacy using eHealth and teamwork in primary healthcare: protocol for a cluster randomised controlled trial.	Protocol	RNs caring for PwO	General practice	Australia
	2022 [45]	Preventing chronic disease in overweight and obese patients with low health literacy using eHealth and teamwork in primary healthcare (HeLP-GP): a cluster randomised controlled trial.	RCT	RNs caring for PwO	General practice	Australia
	2024 [44]	Exploring organisational readiness to implement a preventive intervention in Australian general practice for overweight and obese patients: key learnings from the HeLP-GP trial.	Qualitative	RNs caring for PwO	General practice	Australia
Shaji et al. (2023) [47]		Effectiveness of nurse-led lifestyle modification intervention on obesity among young women in India.	Quantitative	RN caring for PwO	General practice office and telehealth	India
Virtanen (2021) [48]		The impact of lifestyle counselling on weight management and quality of life among working-age females.	Quantitative cohort study	RNs caring for PwO	Primary healthcare	Finland

## Data Availability

The Appendix A include all scoping review search terms, strategies and descriptive characteristics of included studies.

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
