# Peer review of "Nurses’ Role in Obesity Management in Adults in Primary Healthcare Settings Worldwide: A Scoping Review"

_healthcare, 2024, doi:10.3390/healthcare12171700_

Round 1

Reviewer 1 Report

Comments and Suggestions for Authors

This scoping review is superior to other existing reviews (especially in design) in the scientific literature and is worth publishing; however, the author should address several issues.

Specific Comments:

Line 17-19: The statement provided ("This review found that nurses' roles primarily focused on lifestyle interventions (mainly nutrition and physical activity), anthropometric measurements, health planning, goal-setting, supportive care, monitoring progress, and arranging follow-up.") is generally appropriate for a scoping review but could be better aligned with the review's purpose, which is to map out existing literature rather than draw definitive conclusions. A revised version could be: "This review identified that the literature on nurses' roles primarily describes their focus on lifestyle interventions (mainly nutrition and physical activity), anthropometric measurements, health planning, goal-setting, supportive care, monitoring progress, and arranging follow-up."[1].

Line 29-30: Please mention other indexes for categorizing body weight, as this index, while useful, is not without faults[2].

Line 33-35: The term "prevalence" here is misleading. Please revise.

Line 38-39: Please provide studies that demonstrate causality or revise the statement accordingly.

Line 75-77: Support this claim with appropriate references.

Line 100-101: Clarify why a time restriction to the last 5 years was used.

Line 117: PubMed is not a database; it is a search engine. Please revise accordingly.

Line 155-158: Refer to the included studies with proper categorization, as epidemiological standards demand.

Line 379-385: Report all limitations of this study, including those related to study design, etc.

References:

Munn, Z., Peters, M.D.J., Stern, C. et al. Systematic review or scoping review? Guidance for authors when choosing between a systematic or scoping review approach. BMC Med Res Methodol 18, 143 (2018). https://doi.org/10.1186/s12874-018-0611-x

Pray R, Riskin S. The History and Faults of the Body Mass Index and Where to Look Next: A Literature Review. Cureus. 2023 Nov 3;15(11)

. doi: 10.7759/cureus.48230. PMID: 38050494; PMCID: PMC10693914

Author Response

Comment 1:

Line 17-19: The statement provided ("This review found that nurses' roles primarily focused on lifestyle interventions (mainly nutrition and physical activity), anthropometric measurements, health planning, goal-setting, supportive care, monitoring progress, and arranging follow-up.") is generally appropriate for a scoping review but could be better aligned with the review's purpose, which is to map out existing literature rather than draw definitive conclusions. A revised version could be: "This review identified that the literature on nurses' roles primarily describes their focus on lifestyle interventions (mainly nutrition and physical activity), anthropometric measurements, health planning, goal-setting, supportive care, monitoring progress, and arranging follow-up."[1].

We have revised the statement to align with the Reviewer’s recommendation.

Comment 2:

Line 29-30 (now 30-33): Please mention other indexes for categorizing body weight, as this index, while useful, is not without faults[2]. Thank you for the 2023 reference! We have added:

Of note is that the WHO classification system has limitations, such as its inability to accurately measure body fatness associated with health risks. More promising anthropomorphic indices, such as the body shape index, consider fat distribution associated with increased cardiometabolic health risks [3].

Based on Reviewer 1’s recommendations, citation 3 is now: Pray R, Riskin S. The History and Faults of the Body Mass Index and Where to Look Next: A Literature Review. Cureus. 2023 Nov 3;15(11) . doi: 10.7759/cureus.48230. PMID: 38050494; PMCID: PMC10693914

Comment 3:

Line 33-35 (now line 38): The term "prevalence" here is misleading. Please revise. We have changed this statement to:  Obesity is found to affect every single country across the world, and the number of people living with obesity is projected to rise from one billion people affected worldwide today to approximately two billion by 2035 [7]

Comment 4:

Line 38-39 (now lines 43-44): Please provide studies that demonstrate causality or revise the statement accordingly. We have changed this sentence to: Potential psychological complications may include depression, anxiety, self-esteem and poor body image [13,14].

Comment 5:

Line 75-77 (now lines 80-84): Support this claim with appropriate references. We have revised this section to the following:

A previous review that focused on the RN’s role in primary healthcare obesity management for adults (i.e., 19-64 years of age) was published in 2008 [28]. Since then, there have been updated nurses’ clinical practice approaches to obesity pathophysiology and treatment approaches, such as pharmacotherapy (Rust, Prior & Stec, 2020). Given the advances in obesity management and updated clinical guidelines, the aim of this research was to synthesise the current literature in an updated scoping review addressing RNs’ roles in adult obesity management in primary healthcare settings globally.

Comment 6:

Line 100-101 (now lines 105-107): Clarify why a time restriction to the last 5 years was used. We have added: Given nurses’ access to new clinical guidelines for obesity management, a decision was made to limit the search to the past 5 years. Due to an unexpected time lag in completing this review an additional search was conducted to include documents up to April 30, 2024

Comment 7:

Line 117 (now lines 122-123): PubMed is not a database; it is a search engine. Please revise accordingly.

We have changed to: The keywords from the research question were used to construct an initial search strategy beginning with the CINAHL database and the PubMed search engine.

Comment 8:

Line 155-158 (now lines 163-165): Refer to the included studies with proper categorization, as epidemiological standards demand.

We used authors’ descriptions of the methodologies from the included documents, rather than categorize them ourselves. We added the sentence: The documents vary in methodology and are categorized according to the authors’ methodological descriptions as:

Comment 9:

Line 379-385 (Now lines 403-410): Report all limitations of this study, including those related to study design, etc. We have added more limitations pertaining to study design:

A decision was made to conduct a scoping review, given the exploratory nature of the review question and our desire to learn more about gaps between RNs’ current job roles and ongoing knowledge and skills advances in obesity management. The JBI scoping review framework was used to provide rigour and clarity to our process, but scoping review designs tend to be less rigorous than systematic reviews.

It should also be noted that our review did not address differences that may exist within and across countries with respect to nurses’ scope of practice and capacity to regulate themselves as a professional body. (From Reviewer 2)

Comment 10:

References:

 Thank you for these references. We included Pray & Ruskin in the body of the paper and we consulted the paper by Munn et al. to expand our coverage of study limitations in the Limitations section (new lines 388-393).

Munn, Z., Peters, M.D.J., Stern, C. et al. Systematic review or scoping review? Guidance for authors when choosing between a systematic or scoping review approach. BMC Med Res Methodol 18, 143 (2018). https://doi.org/10.1186/s12874-018-0611-x

Pray R, Riskin S. The History and Faults of the Body Mass Index and Where to Look Next: A Literature Review. Cureus. 2023 Nov 3;15(11)

. doi: 10.7759/cureus.48230. PMID: 38050494; PMCID: PMC10693914

Reviewer 2 Report

Comments and Suggestions for Authors

The manuscript entitled ‘’ Nurses’ role in obesity management in adults in primary health care settings worldwide: A scoping review’’ explores the role of nurses in managing obesity among adults within primary health care settings worldwide. Obesity is recognized as a chronic, complex health condition with significant physical and psychological impacts. The World Health Organization (WHO) emphasizes integrating obesity management into existing chronic disease programs within primary care services.

The authors utilized the Johanna Briggs Institute (JBI) methodology to examine the roles of registered nurses in obesity management. The thematic analysis identified six key roles for nurses in this context.

Based in the analyses of the data, the authors concluded that nurses' roles in obesity management primarily involve lifestyle interventions, focusing on nutrition and physical activity, along with conducting anthropometric measurements and health planning. Nurses also play a key role in goal-setting, providing supportive care, monitoring patient progress, and coordinating follow-up care to ensure the effectiveness of obesity management strategies in primary healthcare settings.

It is well written and discussed according to the recent papers in the literature, and presents relevant questions and is of interest for healthcare practitioners who are involved in developing and implementing strategies for obesity management within primary healthcare settings.

Specific points:

·       It would be more informative to discuss how the specific role of nurses in obesity management varies depending on the country and healthcare system. In some regions, nurses may have more autonomy and play a more central role in managing chronic conditions like obesity, while in others, their role may be more supportive.

·       As the authors pointed out that one study [ref: 85] found that 39% of respondents were predominantly relying on common knowledge while providing obesity care. Could you please provide suggestions and actions to address this issue? For instance, participating in research collecting data, and applying evidence-based practices in their care. Also continuous education programs with recent research findings and guidelines in the context of obesity management might improve RNs knowledge and practices.

Author Response

Specific points:

Comment 1:

It would be more informative to discuss how the specific role of nurses in obesity management varies depending on the country and healthcare system. In some regions, nurses may have more autonomy and play a more central role in managing chronic conditions like obesity, while in others, their role may be more supportive.

We have acknowledged this point in our Limitations section (lines 408-410). It should also be noted that our review did not address differences that may exist within and across countries with respect to nurses’ scope of practice and capacity to regulate themselves as a professional body.

Comment 2:

As the authors pointed out that one study [ref: 85] found that 39% of respondents were predominantly relying on common knowledge while providing obesity care. Could you please provide suggestions and actions to address this issue? For instance, participating in research collecting data, and applying evidence-based practices in their care. Also continuous education programs with recent research findings and guidelines in the context of obesity management might improve RNs knowledge and practices.

We have added this content and other references to the Discussion section: More research is needed to determine if barriers associated with RN obesity management require extra education or preparation (pre-and/or post-graduation) to deliver quality care for adults living with obesity and associated physical and mental co-morbidities. In baccalaureate nursing education, content on weight bias and stigma is being included in theory, simulation and practice-based education with promising outcomes (ie., increased awareness of weight bias and strategies for addressing bias) [71,72]. Post-education, RNs can act as positive role models and mentors in situations where obesity bias is present but not acknowledged or addressed. For example, in one cross-sectional study with RNs, almost 50% of participants reported witnessing or providing decreased quality of care of patients with obesity. A primary reason for substandard care was belief that individuals with obesity should do more to help themselves [73]. Proactive educational interventions, particularly in baccalaureate programs, hold promise with respect to new graduates’ more objective assessments and less subjective assumptions about patients with obesity. 

We also added the following sentence to lines 377-378: Pre-and post-education of obesity management for RNs and other healthcare providers can be aided by evidence-based tools